# Durability of Biodegradable Polymer Nanocomposites

**DOI:** 10.3390/polym13193375

**Published:** 2021-09-30

**Authors:** Tatjana Glaskova-Kuzmina, Olesja Starkova, Sergejs Gaidukovs, Oskars Platnieks, Gerda Gaidukova

**Affiliations:** 1Institute for Mechanics of Materials, University of Latvia, LV-1004 Riga, Latvia; olesja.starkova@lu.lv; 2Institute of Polymer Materials, Faculty of Materials Science and Applied Chemistry, Riga Technical University, P.Valdena 3/7, LV-1048 Riga, Latvia; sergejs.gaidukovs@rtu.lv (S.G.); oplatnieks@gmail.com (O.P.); 3Latvian Maritime Academy, Flotes 3-7, LV-1016 Riga, Latvia; gerda.gaidukova@rtu.lv

**Keywords:** biodegradable polymers, nanocomposites, durability, biodegradation, environmental ageing, creep, modelling

## Abstract

Biodegradable polymers (BP) are often regarded as the materials of the future, which address the rising environmental concerns. The advancement of biorefineries and sustainable technologies has yielded various BP with excellent properties comparable to commodity plastics. Water resistance, high dimensional stability, processability and excellent physicochemical properties limit the reviewed materials to biodegradable polyesters and modified compositions of starch and cellulose, both known for their abundance and relatively low price. The addition of different nanofillers and preparation of polymer nanocomposites can effectively improve BP with controlled functional properties and change the rate of degradation. The lack of data on the durability of biodegradable polymer nanocomposites (BPN) has been the motivation for the current review that summarizes recent literature data on environmental ageing of BPN and the role of nanofillers, their basic engineering properties and potential applications. Various durability tests discussed thermal ageing, photo-oxidative ageing, water absorption, hygrothermal ageing and creep testing. It was discussed that incorporating nanofillers into BP could attenuate the loss of mechanical properties and improve durability. Although, in the case of poor dispersion, the addition of the nanofillers can lead to even faster degradation, depending on the structural integrity and the state of interfacial adhesion. Selected models that describe the durability performance of BPN were considered in the review. These can be applied as a practical tool to design BPN with tailored property degradationand durability.

## 1. Introduction

With an increasing global awareness of plastic wastes, there is a huge demand for environmentally friendly solutions such as biodegradable polymers (BP) [1,2]. Moreover, the development of alternative biodegradable materials is motivated due to reasonable limits and the depletion of petroleum resources and rising concerns over the increasing fossil CO_2_ contents in the atmosphere [3]. Recently, many efforts are being made to improve these materials’ quality and functionality, resulting in their applicability in food packaging, agriculture, furniture, construction, engineering and various smart applications [4,5,6,7]. The investigation of degradational processes of polymers and the ways to stabilize them is an extremely important area from the scientific and industrial point of view, and a better understanding of polymer degradation will ensure the long life of the products [8]. Therefore, considering the long-term aspects of such applications, the durability of biodegradable polymers and composites becomes crucial and should be investigated. Moreover, insufficient knowledge of mechanical properties, durability and long-term performance under environmental ageing restricts this new class of sustainable materials for advanced applications [9,10,11].

Generally, bioplastics could be classified into petroleum-based biodegradable polymers, renewable resource-based polymers and polymers from mixed sources (bio- and petroleum-based) as shown in Figure 1. According to the classification, the biodegradability of the polymers depends on the structure but not on the raw material source [12]. Therefore, biodegradable polymers may include both petroleum-based and bio-based polymers. We focus only on several cheap, abundant biodegradable biopolymers herein—polylactide (PLA), polycaprolactone (PCL), polybutylene succinate (PBS), polybutylene adipate-terephtalate (PBAT), polyhydroxyalkanoate (PHA) and thermoplastic starch (TPS).

The addition of nanofillers into BP effectively develops durable bioplastics with controlled functional properties and degradation rates [4,5,6,10,11,12,13,14,15]. Besides improvements in thermal, mechanical and barrier properties, some nanofillers can provide additional functionality to the polymer matrix, e.g., antimicrobial [16,17] and “smart” properties [18,19,20]. The main engineering properties of biodegradable polymer nanocomposites (BPN) were summarized in several recent reviews [21,22,23,24,25]. Some issues related to composites’ preparation and mechanical behaviour with nano-sized reinforcement (i.e., silver nanoparticles, carbon nanofillers, nano-hydroxyapatite and cellulose nanocrystals) in comparison with composites with larger micron-sized inclusions were highlighted in [21]. The results on design, preparation and characterization of biodegradable polymer/layered silicate nanocomposites were reviewed in [22,23]. A comprehensive review of nanocellulose addition’s impact on various synthetic and biopolymer composite materials was provided in [24]. Different properties and potential applications of bio-based poly(butylene succinate) (PBS) composites, including nanocomposites, were highlighted in [25].

Despite increasing interest in the research of BPN, most studies are based on their preparation techniques and the characterization of their fundamental structure–property relationships, while durability issues are rarely reported. The lack of research on the durability of bio-based and biodegradable polymers and composites and the emphasis on the need for this type of research was highlighted in several articles [8,26,27]. Thus, the potential of BPN under different environmental conditions should be thoroughly reviewed and understood to expand their applications to long-term and advanced solutions.

The main aim of the work is to provide insights into the BPN durability and estimate the role of different nanofillers on the overall performance and durability of BP. Recent literature results on the durability performance of BP and BPN were analyzed under environmental ageing and mechanical load conditions. Some existing models for BPN durability prediction were reviewed and discussed.

## 2. Biodegradable Polymers and their Basic Engineering Properties

Biodegradable polymers are abundant and obtainable from natural sources like cellulose, starch and chitosan. They have seen relative success in their applications, but they cannot replace the complete functionality of common fossil plastics like polyolefins, polystyrene, polyethylene terephthalate and others. Thus, commercial biodegradable alternatives to commodity plastics based on polyester structure have been developed and commercialized in the last decade [28]. Emerging bio-based and biodegradable synthetic plastics include polylactic acid (PLA), polycaprolactone (PCL), polybutylene succinate (PBS), polybutylene succinate adipate (PBSA) and polybutylene adipate terephthalate (PBAT). Polyester-produced microorganisms are known as polyhydroxyalkanoates (PHA), which can be further divided into polymer grades like polyhydroxybutyrate (PHB), polyhydroxy valerate (PHV) or their copolymer PHBV. In addition, few conventional fossil-based polymers like polyvinyl alcohol (PVOH) can biodegrade and should be included in this group of materials [29]. To achieve sustainability goals of reducing fossil CO_2_ new more efficient bio-synthesis routes still need to be explored and optimized. In this regard, many advances have been made in the biorefinery field that has yielded most of modern bio-based plastics, but still, issues like relatively higher price and lack of legislation have delayed the transition to bio-based and biodegradable polymer materials worldwide [30].

Biopolymer materials could be differentiated depending on their interaction with water and structure as more hydrophilic and hydrophobic groups are present in the backbone. Natural bio-based polymers are usually hydrophilic; thus, their broad engineering applications are limited, while chemical modifications can change this property resulting in structures like cellulose acetate, nitrocellulose, etc. Thus, biopolyesters have emerged as a non-polar alternative for various applications that require contact with water, humidity and preservation of a sterile environment [31]. In addition, these polymers are often more thermally stable, melt-processed and easily modified with flame retardants for safety purposes [32]. While key characteristics of polymers are achieved with relatively high molecular weight. Especially polymers with molecular weight above 100 000 g/mol can have properties like high ductility, superelasticity and shape memory [33,34].

The characteristic properties of some of the most widely used biodegradable polymers are summarized in Table 1. Starch in the form of thermoplastic starch (TPS) is widely used for various packaging materials and other short life span products. TPS is obtained as a blend using plasticizers and/or other biodegradable polymers as additives. Thus, there is a significant disparity of properties for TPS, but the material itself is usually much more sensitive to water than biodegradable plastics.

Synthetic biopolymers can be sorted into two groups: relatively soft with large elongation values like PBS, PCL, PBAT and the second group with PLA and PHA with relatively high elastic modulus and low elongation values lead to brittleness without additives. PCL has a relatively low melting temperature, limiting its applications and is commonly used for specific purposes like biomedicine. PBS and PBAT have great potential for film preparation required in packaging and agriculture [35,41]. In addition, they are an excellent matrix for the preparation of composite materials. Usually, incorporated particles in the matrix restrict polymer chain movements, resulting in elongation values, which are already low for PLA and PHA. The addition of plasticizers is common for PLA and PHA composite materials [42,43]. Studies indicate that PHA can degrade in various environments, including seawater, while PLA needs specific soil conditions [44]. The drawback of PHA is the relatively high cost of production. In addition, PHA and PLA have a relatively narrow range of thermal processing, while PBS and PBAT have been reported to be much more stable during melt processing [45,46].

## 3. Potential Nanofillers for Biodegradable Polymers

The main drawback of biopolymers is that most of them have poor mechanical and thermal properties limiting their use in structural applications. Both natural and synthetic nanofillers could be used to improve the physical-mechanical properties of biopolymers. In the case of both matrix and filler derived from renewable resources, a fully renewable and biodegradable nanocomposite could be produced [39].

Different nanofillers may introduce different properties to BPN resulting in specific applications [12]. Mostly, recent applications of BPN are limited to packaging, biomedical, antibacterial and smart applications. The scope of possible applications for different combinations of BP and nanofillers is reviewed in Table 2. Examples of smart applications of BPN include piezoresistive vapour sensors for PLA filled with multiwall carbon nanotubes (MWCNT), shape-memory applications for poly(d,l-lactide) filled with Fe_3_O_4_, and electrical/electromagnetic applications for PLA/PHBV filled with MWCNT [18,19,20]. The addition of electrically conductive fillers (e.g., carbon black, carbon nanotubes, nanofibres, graphene, Fe_3_O_4_) into biopolymers may result not only in improved nucleating, mechanical, thermal and fire-retardant properties, but also may introduce tailored electrical and thermal conductivity [15,47]. These composites can be promising as materials for manufacturing sensors with sensitivity to such factors as strain, temperature or organic solvents [7].

It should be noted that the addition of nanofillers could negatively affect biopolymer properties. For example, the advanced degradation of PLA chains resulting in reduced thermomechanical properties was observed upon the addition of some metal oxides such as calcium oxide (CaO), magnesium oxide (MgO) or other metallic compounds such as layered double hydroxides [47]. Similarly, the addition of untreated ZnO nanoparticles into PLA resulted in intense degradation at melt-processing temperature, described by the transesterification reactions and ‘unzipping’ depolymerization of PLA [11,26]. Nevertheless, the surface treatment of ZnO by using silanes may improve the physicochemical characteristics of PLA.

## 4. Biodegradation of BPN

The overall degradation process of biopolymers and biocomposites could be related to light, heat, moisture, chemical and microbial treatment on the bulk polymer material [60]. Biodegradation (i.e., biotic degradation) is a chemical degradation of materials (polymers) provoked by the action of microorganisms such as bacteria, fungi and algae. While a biodegradable polymer is a degradable polymer wherein the primary degradation mechanism is through the action of metabolism by microorganisms [61]. Different bacteria mainly guide the biodegradation of a macromolecular structure. Commonly, applying the complex factors of light, heat and microorganisms could significantly pronounce the intensity of polymers’ physical and chemical changes, leading to a noticeable drop in the material’s properties, partial disintegration and complete disappearance. For the efficient biological activity of bacteria, the polymer materials should have at least contact with soil and compost. In contrast, the full burial in the soil media of the polymer could facilitate the biodegradation process. In general, all biological functions, for example, bacterial biodegradation, are strongly dependent on the presence of water [62].

The biodegradation of polymer material could temporarily or permanently create small molecules that should be accumulated in the environment [63]. As reported, the formed oligomers, monomers and metabolic intermediates can interact with living organisms in the soil, adversely affecting the environment [64]. To that, environmental issues of the persistency and ecotoxicity of the developed compounds become very important in the biodegradation process investigations [65].

Many authors report that the polymer chain topology, macromolecular network structure, molecular chain weight and size can severely affect PBS, PBSA, PLA, PHA and other bio-based polymers biodegradation in soil [66,67,68]. The temperature, moisture, pH and the population of active microorganisms are essential factors to facilitate the polymers’ biodegradation [69]. These conditions are broadly reviewed and reported in the literature; they depend on the soil characteristics, which vary from place to place and season to season. In comparison, the industrial composting conditions are easy to control due to several strictly physically/chemically controlled parameters and the standardized environment [70,71].

Several authors report that the biodegradation of the composites differs from the unfilled polymers [55,72,73] (e.g., see Figure 2). The degradation process depends on the nature, the chemical modification and the content of the used fillers [74,75]. Synthetic and natural fillers of different sizes and shapes are broadly used to control the biodegradable polymers performance properties [55,72,76]. Carbon, metallic, metallic oxide, cellulose and other micro-and nanoparticles have been very popular in the last decade [77,78,79,80]. Fillers could enormously change the overall degradation characteristics of the polymer materials [81]. For example, spent coffee particles significantly enhanced the tensile properties but strongly decreased the biodegradation time for biopolymers [82]. Similar biodegradation enhancement in the soil is observed for microcellulose and nanocellulose particles loaded biocomposites [83].

The changed biodegradation mechanism of the biocomposites due to the nanoparticles’ antibacterial properties was reported in [84,85]. It is possible to improve the antibacterial properties of bio-based polymers by adding the nanofillers having antibacterial properties (e.g., ZnO, Ag, MMT, etc.) [1,16,86]. Thus, the antimicrobial activity of ZnO-modified PBS films was proven to be effective against representative food spoilage bacteria (*S. aureus* and *E. coli*) at minimal content of 6 wt.% of ZnO [16]. Moreover, a synergistic effect in enhancing the antimicrobial properties against the bacteria, as mentioned above, was found by combinatorial use of Ag/ZnO/CuO nanofillers in the formulation of starch-based films [56].

The microorganisms multiply and prosper at mild temperatures in the presence of moisture and a source of carbon [54]. There is a significant concern to add antimicrobial properties to bio-based polymers to diminish the quantity and propagation of microbes (bacteria, fungi) by using antimicrobial agents. The antibacterial activity is analyzed mostly by transmission electron microscopy (TEM), scanning electron microscopy (SEM), Fourier-transform infrared spectroscopy (FTIR), nuclear magnetic resonance (NMR), zeta potential and dynamic light scattering (DLS) analyses [87].

Another way is to add the antibacterial agents such as, e.g., the bacteriocin (antibacterial peptides) to crystalline nanocellulose and incorporate such bacteriocin immobilized crystalline nanocellulose into bio-based polymers as antibacterial agents to have antibacterial properties with enhanced strength of the films and better biodegradability [88].

Still, the antimicrobial/antibacterial action mechanisms when the nanofillers are added to biopolymers are not fully understood [16,17,47,48]. However, the leading hypothesis is related to the photocatalytic generation of many reactive oxygen species to the formation of the ions [17], consequent leakage of intracellular substances, and lastly, the destruction of bacterial cells [16].

## 5. Durability Performance of BPN

According to a general definition provided in [9], material durability is related to the ability of a material to withstand a wide variety of physical processes and chemical degradation reactions from the exterior environment. The environmental factors can be solitary or combined action of moisture, oxygen and bacteria attacks, mechanical loading, wear and tear, and extreme temperature conditions. Basic durability tests include thermo- and photo-oxidative ageing, creep and fatigue, water absorption and hydrothermal ageing (Figure 3). Due to the breakdown of the macromolecules’ structure from the water absorption and oxidation process–induced during the exposure to the environments, the functional properties of biopolymers and bio-based nanocomposites could deteriorate. For example, in [9] it was shown that the applied accelerated weather conditions did not cause significant changes in the mechanical properties of biocomposites made of flax fibres and epoxidized soybean oil–based thermosetting resin. An increase of hardness, tensile strength and modulus, and decrease of elongation at break and impact strength was attributed to the decreased chain mobility and increased crosslinking density after the tests.

Due to environmental degradation, different reversible and irreversible consequences may occur to the tested materials, such as decreased molecular weight (chain scissoring), reduced mechanical properties, embrittlement and cracks, colour fading and spots [1,8,89]. Moreover, it should be noted that the ageing behaviour and mechanism of the unfilled polymers are usually less complex than those for filled composite materials. This is due to the presence of different components in the composite such as fillers, fibres, additives, plasticizers, antioxidants, etc., each contributing to the environmental degradation of the composite as a whole [90,91].

Usually, ageing tests are rather long-term, lasting several years or even decades, and, therefore, accelerated ageing tests are applied to imitate specific environmental conditions at an increased rate. Moreover, these studies allow predicting the performance and investigate the degradation mechanisms of the materials and are very important to understand the material ageing behaviour for specific conditions and applications [92,93,94,95].

Environmental ageing in some cases causes an increase in the degree of crystallinity of the polymers. For instance, the rise in the degree of crystallinity by 50% was found for the neat PLA after accelerated weathering [27]. It was attributed to the relief of thermal stresses introduced due to the manufacturing process, which occurs under high temperature, and also re-aligning of broken chains due to chain scissoring into a more organized structure.

Ageing tests include different conditioning, e.g., in a climate chamber, exposure of natural weathering and UV irradiation, photo-oxidation, thermal-oxidation, water absorption, humidity, microbial, chemical degradation, thermal cycling/fatigue or a combination of these methods. The results of recent studies of the durability of BPN are summarized in Table 3.

### 5.1. Thermo-Oxidative Ageing

Generally, thermal degradation of polymers is a very complex phenomenon that involves physical, chemical and thermal processes [112]. During the manufacturing process and service life, polymers are generally exposed to thermo-oxidative degradation, which causes degradation of their performance, especially for long-term applications [90,113,114]. According to Table 3, the thermal degradation of PLA filled with different nanofillers (ZnO and CNT) caused the change in glass transition [26] and crystallization [15] temperatures. Similar results regarding the change in crystallization temperature were reported for PBS filled with CNF [55] and PHB filled with bentonite [103], and regarding the change in glass transition temperature due to thermal degradation for starch-filled with MWCNT [102], PHB filled with MMT [87] and PCL filled with nanoclay [105].

As reported in [115], for BP at elevated temperatures (above glass transition temperatures) a random chain scission mechanism occurs, determining a significant level of molecular degradation and polymer embrittlement. In addition, it was experimentally proven [116] that the oxidative degradation of PLA occurs at moderate temperatures (below PLA melting temperature) with a significant reduction of the polymer molar mass. According to Figure 4 the molar weight of PLA, aged at different constant temperatures (100, 130 and 150 °C), changed almost linearly as a function of temperature at different time-sections (indicated on the graph). An antioxidative degradational process could be minimized by adding the antioxidants to polymers, such as hindered phenols or amines and organophosphorus compounds [117].

To study the kinetics of thermal degradation of BP and BPN different isoconversional methods could be applied [101,109,118,119,120]. The degradation rate for the isothermal process is given by a general relationship [101]:(1)$$\frac{d\alpha }{dt}=k\left(T\right)f\left(\alpha \right),$$
where $k\left(T\right)$ is the rate constant at temperature *T*, α is a specific degree of degradation or conversion (e.g., given by the mass loss in TGA tests) and $f\left(\alpha \right)$ is a function of the reaction model related to the degradation mechanism.

For non-isothermal measurements at a constant heating rate $\beta =\frac{dT}{dt}$ and the rate constant given by Arrhenius equation, Equation (1) takes the form:(2)$$\beta \frac{d\alpha }{dT}=A\exp\left(\frac{E_{a}}{RT}\right)f\left(\alpha \right),$$
where *A* is a constant, *E_a_* is the activation energy and *R* is the gas constant.

Equation (1) is the basic equation used for the prediction of the degradation evolution. In general, determining the pre-exponential factor and the activation energy is challenging since both parameters could be interrelated conversion functions. The isoconversional methods provide simplified procedures for characterizing the degradation kinetics by presuming temperature independence of the pre-exponential factor and the activation energy in Equation (1). The latter could be evaluated without presuming any specific form of the degradation function $f\left(\alpha \right)$, while changes in *E_a_* vs. α changes are assumed to be related to changes in the degradation mechanism. Isoconversional methods require a series of experiments with different temperature programs and obtaining *E_a_* as a function of the conversion degree [118].

The activation energy can be calculated by various methods. Friedman’s method is based on Equation (2)[101,109,118]:(3)$$\ln\frac{d\alpha }{dT}=\ln\frac{A}{\beta }+\ln f\left(\alpha \right)-\frac{E_{a}}{RT},$$

It is seen from Equation (2) that if the function $f\left(\alpha \right)$ is constant for a particular value of α, the sum of the first two terms in Equation (3) also give a constant. Then, plotting $\ln\left(d\alpha /dT\right)$ vs. 1/*T* give straight lines with the slope $\left(-E_{a}/R\right)$. 

In the Ozawa–Flynn–Wall method [109,118,120], it is assumed that the conversion function $f\left(\alpha \right)$ is invariant to the heating rate irrespective of the degree of conversion α. Equation (2) could be written as follows:(4)$$\ln\beta =\ln\left[\frac{Af\left(\alpha \right)}{d\alpha /dT}\right]-\frac{E_{a}}{RT}$$

The method involves measuring the temperatures corresponding to fixed values of α from tests at different heating rates β. The activation energy could be determined from the slope of lnβ vs. 1/T straight lines according to Equation (4).

Titania nanoparticles incorporated into PLA/PHBV blends catalyzed the degradation process and inhibited the diffusion of the degradation volatiles out of the sample [118]. TGA tests were performed at different heating rates and the activation energy of degradation was calculated according to the Ozawa–Flynn–Wall model Equation (4). Alternative isoconversional methods for processing thermogravimetric data are highlighted in [101,118,120].

Chrissafis et al. have compared thermal degradation mechanisms of PCL and its nanocomposites containing different nanoparticles (pristine and modified MMT, MWCNT and fumed silica). Thermogravimetric analysis using non-isothermal conditions was performed at different heating rates and the activation energies were estimated using the Ozawa–Flynn–Wall Equation (4) and Friedman methods Equation (3). It was verified that nanoparticles did not affect the degradation mechanism but only the decomposition rate and thermal stability of PCL. Accelerated decomposition of PCL was observed for nanocomposites filled with modified MMT with quaternary ammonium salts and SiO_2_ nanoparticles promoted by aminolysis and hydrolytic degradation due to the presence of the reactive groups on their surface. At the same time, unmodified MMT and MWCNT inhibited thermal degradation of PCL due to the shielding effect.

Nanoreinforcing is an effective way to improve the thermal stability of BP extending their high-performance applications. For instance, the results of DSC showed that the presence of MWCNT had a nucleating effect on both the melt crystallization and the cold crystallization of PLA [15]. Similarly, it was proven that ZnO acted as a disruptor of the PLA crystallization process, causing the degradation of PLA polymer chains during melt processing, and shifted the polymer glass transition temperature (*T*_g_) to lower temperatures [26]. Moreover, it was shown that PBS polymer matrix could effectively shield the NFC nanofiller from thermal degradation resulting in a lower mass-loss rate and degradation over a wider and upper-temperature range [55]. Adding cellulose nanofibres to glycerol plasticized starch significantly enhanced the activation energy by 52% [101]. Meanwhile, for PHB/organically modified clay nanocomposites, the activation energy did not vary greatly with the degree of degradation, denoting degradation in one step with similar values for pure PHB and all nanocomposites [103].

### 5.2. Photo-Oxidative Ageing 

Exposure to ultraviolet (UV) light can limit the scope of applications for BP as they can become fragile during storage, transportation and outdoor use [121]. The operational environment causes oxidation and cleavage of small molecular components, which leads to the deterioration of physical properties [122]. The addition of TiO_2_ nanofillers improves UV resistance and the mechanical performance of BP and conventional petroleum-based polymers [100]. According to FTIR results indicated the degradation of the poly(butylene succinate-co-butylene adipate) (PBSA) matrix caused by high-energy UV light was significantly reduced with the addition of only 1.5 wt% of TiO_2_ nanoparticles.

The results obtained on viscosity analysis indicated that TiO_2_ nanoparticles inhibited the chain scission of PBSA matrix under irradiation and led to the reduced deterioration of their mechanical properties than that of unmodified PBSA films during the photoaging process [100]. According to Figure 5, the relative change of complex viscosity of PBSA filled with TiO_2_ nanoparticles, after 360 h of UV irradiation was maximally reduced for PBSA with 1 wt.% of TiO_2_. It can be attributed to the diminished dispersion at higher filler loadings leading to faster degradation, depending on the structural integrity and the state of interfacial adhesion. By using FTIR, pyrolysis gas chromatograph-mass spectrometer (PGC-MS), DSC and SEM, similar results were also obtained for the PBSA matrix filled with ZnO nanoparticles (0.5– wt.%), demonstrating that ZnO nanoparticles can hinder the photodegradation of PBSA [54].

The most appropriate and popular measurement of photodegradation is UV irradiation in a weatherometer [8]. This method allows outdoor accelerated exposure testing of plastics at the simulated desert and sub-tropical climatic conditions and applies to a range of polymer materials including films, sheets, laminates and extruded and moulded samples.

### 5.3. Water Absorption and Hygrothermal Ageing

Moisture or water affects hydrophilic constituents of BPN through immersion, cycles of spraying and condensation [123]. Water transport is governed by three mechanisms, i.e., the diffusion through micro-gaps between polymer chains, capillary transport into interfaces and transport through micro-gaps caused by swelling of hydrophilic constituents [90,124,125].

Some of the BP (e.g., PLA, PVA, starch, cellulose acetate) are well known to be able to absorb a considerable amount of water due to their amorphous nature that allows water molecules to penetrate more easily than into semi-crystalline polymers (e.g., PBS, PCL, etc.) [11,13,96,99]. To minimize the water absorption content and subsequent degradation of physical and mechanical properties of BP different nanofillers, such as ZnO [11] and CNC [96] could be added. The nanofillers act as crosslinking entanglements leading to lower water absorption in the nanocomposite than the neat polymers.

Thus, significant improvement in barrier properties of poly(D,L-lactide) (PDLLA), i.e., water absorption resistance, was obtained by the addition of the CNW (Figure 6) [13]. These results showed that even a small quantity of cellulose nanowhiskers (1 wt.%) inhibited water absorption and hence retarded the degradation, modifying the kinetics of the hydrolytic process in PDLLA polymers.

According to the Nielsen model, the relative permeability coefficient is inverse proportional to the tortuosity factor [23,125]:(5)$$\frac{P}{P_{0}}=\frac{1-\varphi }{k}$$
where *P* and *P*_0_ are the permeabilities of the composite and neat polymer, respectively.

The diffusion phenomena in polymers filled with filler particles could be associated with the tortuosity factor *k,* which is a function of the filler aspect ratio (α) and volume content (φ) [23,125,126,127,128]
(6)$$k=1+\frac{\alpha }{2}\cdot \varphi .$$

For instance, PLA filled with ZnO [11,129], CaO or MgO [47], MMT [17,130], MWCNT [15] and CNW [13] are characterized by different tortuosity factors according to Equation (6). The tortuosity factor–filler volume fraction relationship is shown in Figure 6. The densities, relative weight fractions and aspect ratios for the nanofillers were taken from the data provided in the according papers.

From Figure 7 it is obvious that the tortuosity factor for 1D nanoparticles ZnO, MgO and CaO is close to unity, while 2D CNW only slightly contributes to its increasing at high filler volume fractions. MMT and MWCNT having a high aspect ratio (50 and 100, accordingly) improve the barrier properties of the polymers [90]. Hence, inhibited water absorption can retard the degradation, modifying the kinetics of the hydrolytic process in BP. Moreover, a good filler–matrix adhesion would reduce water molecules’ penetration into BP to reduce the water absorption properties [131,132].

Bharadwaj [133] has modelled permeability in polymer-layered silicate nanocomposites and modified Nielsen’s model by taking into account orientational effects. The Bharadwaj model is given by the following relation [99,133]:(7)$$\frac{P}{P_{0}}=\frac{1-\varphi }{1+\frac{\alpha \varphi }{2}\cdot \frac{2}{3}\cdot \left(S+\frac{1}{2}\right)}$$
where *S* is the orientation of fillers in the nanocomposites. *S* take the values of −0.5, 0 and 1 for fillers oriented perpendicularly, randomly and parallel to the membrane surface. It is seen that the Bharadwaj model (Equation (7)) reduces to Nielsen’s model (Equation (5)) at *S* = 1.

Okamoto highlighted that relative permeability as a function is inverse proportional to the tortuosity factor for different biodegradable polymer/layered silicate nanocomposites [23].

The water absorption property of BP and BPN can also be determined by Fick’s law that in some cases could be given by a simplified equation [131]:(8)$$\frac{M_{t}}{M_{s}}=k\cdot t^{n}$$
where *M_t_* is the moisture content at time *t*, *M_s_* is the moisture content at the saturated point and *n* are constants determined from the fitting curve of plot log(*M_t_*/*M_s_*) vs. log(*t*), accordingly. Thus, depending on the *n*-value, the moisture diffusion property of the composite can be divided into three cases: when *n* = 0.5 and the diffusion is Fickian, when 0.5 < *n* < 1, and the diffusion is non-Fickian or anomalous; and when *n* > 1 [132].

Another parameter of Fick’s model is the diffusion coefficient (*D*) which determines the ability of water molecules to diffuse and penetrate the composite structure. Its value is calculated from the slope of the plot of *M*_t_/*M*_s_ vs. time (*t*^0.5^) by the following equation [90,131,134]:(9)$$\frac{M_{t}}{M_{s}}=\left(\frac{4}{h}\right){\left(\frac{D}{\pi }\right)}^{0.5}\cdot t^{0.5}$$
where *h* is the thickness of the specimen.

Cosquer et al. have studied the influence of graphene nanoplatelets (GnP) on water absorption kinetics of biodegradable PBS [99]. GnP, being hydrophobic nanofillers of a high aspect ratio, act as efficient impermeable barriers. The diffusivity of PBS decreased by about 40% compared to nanocomposites with 2wt.% GnP (Figure 8). The improvement was attributed to a purely geometric type phenomenon, i.e., with increasing the tortuosity. The tortuosity factor was estimated by the ratio of the diffusion coefficients of the neat polymer and nanocomposites by using a relation similar to Equation (5). The tortuosity factor was found to be independent of the water activity. The Bharadwaj model Equation (7) applied for fitting the experimental data on water and dioxygen permeability showed reasonable results.

Hydrothermal ageing results in changes in physical (e.g., *T*_g_) and mechanical properties. The inherent structure is deteriorated appearing in loss of interfacial adhesion and reinforcement efficiency. Thus, the durability of composites and the extent of degradation could be assessed by comparing these parameters in the reference and aged states.

The interfacial adhesion between the filler and polymer matrix plays an essential role in determining the mechanical properties of composites. Pukanszky’s model is among the most widely used models for assessing the filler–matrix bond strength. Originally, the model was developed for particulate polyolefin-based composites [135,136], although later it has been successfully applied for other heterogeneous polymer systems, including bio-based and biodegradable blends [137] and nanocomposites [98]. According to Pukanszky’s model, the composite strength $\sigma_{c}$ and the polymer matrix strength $\sigma_{m}$ are related by the equation [135,136,137,138]:(10)$$\sigma_{c}=\sigma_{m}\frac{1-\varphi }{1+2.5\varphi }\exp\left(B\varphi \right)$$
where *B* is the adhesion parameter: an empirical constant, which is dependent on the surface area of the particles, particles density and interfacial bonding energy. *B* value is 0 for very weak adhesion and can be increased, depending on the adhesion strength.

The interfacial adhesion of PLA filled with different types of fumed silica nanoparticles was estimated by Dorigato et al. [98]. The adhesion parameter *B* was dependent on the surface treatment of SiO_2_ and varied from 3.8 to 2.5 with the highest value for pristine nanoparticles. An opposite effect of the improved interfacial adhesion with surface modifications of the filler is observed for PLA/sugarcane leaves fibre biofilms [131]. Bleaching treatment by H_2_O_2_ improved the interfacial adhesion between PLA and sugarcane leaves and thus enhanced biofilms’ tensile strength, evidenced by the increased B factor from 6.6 to 7.5. Low adhesion factors of around 1.3 were found for PBS/wine lees [110] and PBS filled with microcrystalline cellulose particles [138]. The filler–polymer bond strength was enhanced by chemical modifications of the MCC surface [138]. Nanni and Messori have studied the strength properties of PHBH and PHBV composites filled with natural fillers [139]: the determined B values were in the range of 2.2–3.3 for pristine and 2.7–3.6 silane-treated fillers.

The interfacial adhesion is deteriorated due to ageing. By comparing *B* factors for pristine (unaged) and aged composites it is possible to estimate the extent of degradation on the mechanical properties quantitatively. For example, hydrothermal ageing of PBS/MCC composites [138] decreased the filler–matrix bond strength manifested in B drop from 1.37 to 0.78 for the reference and aged samples, respectively. Sugiman et al. reported about 1.4-fold decrease of B caused by water absorption in a polymer system filled with inorganic fillers [140].

The reinforcement efficiency is also affected by ageing. Nanofillers could act as reinforcement and contribute to the improvement of elastic properties of polymers. The reinforcement efficiency could be estimated in different ways. In a general case, the overall effectiveness of the reinforcement in a composite could be estimated by a simple empirical relationship [141,142,143]:(11)$$E_{c}=E_{m}\left(1+r\varphi \right)$$
where *r* is the reinforcement efficiency factor; $\prime E_{c}$ and $E_{m}$ are the elastic moduli of the composite and matrix, respectively. Platnieks et al. have studied the elastic properties of PBS/NFC composites processed by melt blending and solution casting [142]. By comparing *r* factors, the authors demonstrated the superior effectiveness of the former processing method. Hydrothermal ageing effects on the stiffness reduction of epoxy/graphene oxide nanocomposites appeared in the decrease of the reinforcement efficiency and *r* drop from 1.6 to 0.14 were highlighted in [141].

An alternative way to assess the filler contribution into the elastic properties of composites is based on an analysis of DMTA data and elastic moduli evolution when passing the glass transition. A so-called parameter *C*, relating the storage moduli in the glassy $E_{g}^{\prime }$ and rubbery $E_{r}^{\prime }$, is given by the ratio [141,142,143]:(12)$$C=\frac{{\left(E_{g}^{\prime }/E_{r}^{\prime }\right)}_{c}}{{\left(E_{g}^{\prime }/E_{r}^{\prime }\right)}_{m}}$$
where subscripts *c* and *m* correspond to composite and matrix, respectively. For well-dispersed fillers and good compatibility with the polymer matrix, *C* < 1. The lower is *C*, the most efficient the reinforcement effect is. *C* factors of PBS filled with nanofibrillated cellulose (NFC) prepared by different processing routes were compared [142]. It was found that samples with 15 wt.% NFC processed by melt processing are characterized by higher *C* = 0.69 than those processed by solution casting. The reduced reinforcement efficiency and increase of *C* factors related to hydrothermal ageing effects were found in [141] by the example of epoxy/graphene oxide nanocomposites.

### 5.4. Creep

Viscoelastic properties of polymer-based composites have a critical role, especially in long-term applications, indicating the time-dependent deformation of materials as a function of temperature, stress and strain [144,145,146,147]. With increasing stress/temperature values, a creep compliance function becomes nonlinear on stress which can be described by various phenomenological models considering the creep of the composite and neat matrix. Different additives could be introduced to reduce creep deformations [10,98,120,139,145]. Nanofillers reduce the creep curve’s elastic component and the viscous flow of the material with increment in viscoelastic deformations [145].

To estimate the effect of the fillers on the long-term deformability of BP, the creep parameters should be denoted. Figure 9 shows a schematic strain vs. time curve in a standard creep-recovery.

The total creep strain ε is given by a sum of three components:(13)$$\varepsilon =\varepsilon_{el}+\varepsilon_{ve}+\varepsilon_{vp}$$
where subscripts *el*, *ve* and *vp* correspond to elastic, viscoelastic and viscoplastic strain components, respectively. The residual strain ($\varepsilon_{res}$) is defined as a permanent viscoplastic strain accumulated during the whole loading period and remaining after unloading after a time period longer than that of loading.

It was found [10] that the addition of cellulose nanofibrils (CNF) to starch-based nanocomposite films significantly decreased all creep deformations (viscoelastic and plastic, elastic and residual) shown in Figure 10. The concentration of CNF above 20 wt.% was found to accelerate the creep behaviour due to poor dispersion, whereas the nanocomposite films with CNF content of 15 wt.% revealed the lowest creep performance.

The most common model for creep description is the three-parameter Findley power law given by the following equation [110,139,148,149]:(14)$$\varepsilon \left(t\right)=\varepsilon_{el}+kt^{n}$$
where *k* and *n* (0 < *n* < 1) are material parameters. The power-law models are considered empirical models without attaching importance to a physical background. 

The Burgers model is a combination of Maxwell and Kelvin–Voigt elements connected in series. According to Equation (13) and the Burgers model formulation, the creep strain is given by the following relation [10,98,139,150]:(15)$$\varepsilon \left(t\right)=\frac{\sigma }{E_{M}}+\frac{\sigma }{E_{K}}\left[1-\exp\left(-\frac{t}{\tau }\right)\right]+\frac{\sigma }{\eta_{M}}t$$
where *E_M_* and *E_K_* are the elastic moduli of the Maxwell and Kelvin springs, τ = *η_Κ_*/*E_K_* is the retardation time of the Kelvin–Voigt element; *η_Κ_* and *η_Μ_* are the viscosities of the Kelvin and Maxwell dashpots, respectively.

The creep of glassy solids and semicrystalline polymers is described by the Kohlrausch–Williams–Watts (KWW) model. This is based on considerations that “viscoelastic changes in polymeric matrices occur because of incremental molecular jumps due to several segments chains jumps between different positions of relative stability”. The creep strain is given by a Weibull-like function [120]:(16)$$\varepsilon \left(t\right)=\varepsilon_{i}+\varepsilon_{c}\left[1-\exp\left(-{\left(\frac{t}{t_{c}}\right)}^{\beta_{c}}\right)\right]$$
where $\varepsilon_{i}$ is the instantaneous elastic strain, $\varepsilon_{c}$ is the limit viscous creep strain, $t_{c}$ and $\beta_{c}$ are the scale (characteristic time) and shape parameters, respectively. Expanding Equation (16) in a series and considering the first term only derives from the Findley model Equation (14).

The Weibull distribution equation is also applied to model the creep recovery behaviour. The recovery strain $\varepsilon_{rec}$ is determined by the viscoelastic strain recovery $\varepsilon_{c}$ and the residual strain $\varepsilon_{res}$ caused by a viscous flow effect and is given by the following equation [110]:(17)$$\varepsilon_{rec}\left(t\right)=\varepsilon_{c}\left[\exp\left(-{\left(\frac{t-t_{0}}{t_{r}}\right)}^{\beta_{r}}\right)\right]+\varepsilon_{res},$$
where $t_{0}$ is time of stress removal, $t_{r}$ and $\beta_{r}$ are the characteristic time and shape parameters, respectively.

Temperature growth results in accelerating relaxation processes in polymers and thus changing their viscoelastic response (e.g., creep compliance, relaxation modulus). This fact is widely applied to predict the long-term properties of polymers and their composites by using Time–Temperature Superposition Principle (TTSP) [146,147]. TTSP is based on the assumption that time and temperature are interrelated and interequivalent. A temperature increase leads to a parallel shift of the relaxation spectrum of a polymer, and this shift is characterised by so-called shift factors $a_{T}$. The long-term viscoelastic behaviour is predicted by shifting the short-term test data presented in logarithmic time axes to a generalized master curve for $\log a_{T}$ values. The lifetime of a polymer system *t* at an operating temperature *T* is determined by a ratio of the shift factors according to the relation [118]
(18)$$t=\frac{a_{T_{0}}}{a_{T}}\cdot t_{0}$$
where *T*_0_ is the reference temperature, *t*_0_ is the lifetime at *T*_0_; $a_{T_{0}}$ and $a_{T}$ are the shift factors at corresponding temperatures. For simplicity, $a_{T_{0}}=1$ is usually taken. TTSP has temperature limitations in terms of the shift function. The Williams–Landel–Ferry (WLF) equation is valid for the temperature range between *T*_g_ and *T*_g_ +100 °C [146,147]:(19)$$\log a_{T}=-\frac{C_{1}\left(T-T_{0}\right)}{C_{2}+T-T_{0}}\prime$$
where $C_{1}$ and $C_{2}$ are material parameters.

The Arrhenius equation is applied for $a_{T}$ calculations at *T* < *T*_g_:(20)$$\log a_{T}=\frac{E_{a}}{2.303R}\left(\frac{1}{T}-\frac{1}{T_{0}}\right)\prime$$
where *T*_0_ is taken in Kelvin; other designations are the same as in Equation (2).

Nanni and Messori [139] applied the Burger, KWW and Findley models to describe the nonlinear creep of PBS biocomposites filled with wine lees. By comparing parameters of the models representing elastic, viscoelastic and viscoplastic strain components, it has been quantitatively demonstrated that the addition of fillers into PBS resulted in the reduced creep of biocomposites. The earlier authors’ study successfully applied the KWW model to describe creep PHBH- and PHBV-based biocomposites [120]. The master curves were generated by applying TTSP and the temperature shift factors were calculated according to the WLF model Equation (19). The long-term predictions for wine lees–filled biopolymers demonstrated much lower creep in the same time spans.

The creep and creep-recovery behaviour of starch-based nanocomposite films with CNF up to 20 wt.% have been studied by Li et al. [10]. The experimental data were effectively fitted by the Burgers model with parameters strongly dependent on the amount of the filler. TTSP was successfully applied for predicting the long-term creep behaviour of nanocomposites. The temperature shift factors were calculated according to the Arrhenius equation Equation (20).

Amiri et al. applied the Findley model to describe nonlinear creep of bio-based resin (methacrylated epoxidized sucrose soyate (MESS) reinforced with flax fibres [148]. Following TTSP and considering biocomposites as thermorheologically complex materials, the authors used horizontal and vertical shifts to generate the master curves. The long-term prediction agreed well with the experimental data.

In a recent study of Ollier et al. [149], creep of PCL reinforced with pristine and organo-modified bentonites up to 3 wt.% was investigated. The Findley and Burgers models were applied. Master curves were constructed using TTSP demonstrating substantial improvement in the creep resistance of nanocomposites for the long term. 

The improved creep stability of PLA filled with fumed silica nanoparticles of different specific surface areas and surface functionalization was discovered by Dorigato et al. [98]. The authors applied the Burgers model and demonstrated that nanoparticles mainly contribute to the increased values of viscous components (*η_Κ_* and *η_Μ_* in Equation (15)).

Guedes et al. [150] and the Burgers model applied the modified three-element standard solid model of Kontou–Zacharatos [106] to describe the nonlinear viscoelastic behaviour of PLA-PCL fibres monitored in creep, stress-relaxation and quasi-static tensile tests. Dry and saturated in saline solution fibres were tested. Based on phenomenological considerations, the elastic spring is replaced by a nonlinear strain-dependent spring and the linear dashpot in the Maxwell element is replaced by an Eyring type one. Such modifications allowed authors to reduce the required fitting parameters and describe nonlinear viscoelastic–viscoplastic behaviour under moderate and large deformations, both in monotonic and cyclic loading.

Ding and coworkers have studied creep and stress relaxation of PBAT biocomposites containing CNT up to 5 wt.% [110]. The authors applied the Findley and Burgers models for creep description, while creep recovery was modelled by the Weibull distribution function given by Equation (17). PBAT/CNT nanocomposites possessed lower viscoelastic and viscoplastic strains that appeared in changed creep model parameters than the unfilled polymer.

Kontou et al. have studied the time-dependent behaviour of PBAT/PLA blends (commercial name Ecovio^®^) reinforced with different types of wood fibres up to 30 wt.% [106,151]. Highly nonlinear viscoelastic/viscoplastic behaviour observed in creep tests was modelled by the Burgers and Findley models [106]. A constitutive model presenting a combination of the transient network model, related to the viscoelasticity, with a plasticity theory has been developed in [115] describing the experimental data of stress-relaxation, monotonic loading and creep-recovery in a unified manner. 

Qiu et al. have studied the time-dependent plastic failure of PLA/PBS blends in tensile tests at different strain rates [123]. The deformation behaviour of the blends with the improved ductility was modelled by the Chaboche viscoplastic model with nonlinear hardening variables.

### 5.5. Modelling of Mechanical Properties Accompanied by Biodegradation

The structure and properties of biodegradable polymers change in time comparable to non-biodegradable counterparts, with the test time and service life of materials. This fact should be considered when modelling such materials’ time-dependent properties (e.g., creep, stress relaxation, fatigue, etc.). Parameters involved in traditional models need to be related to “inherent” degradation of the structure that could also be accelerated by an external mechanical load. In several studies for different biopolymers [148,152,153,154,155,156], degradation of mechanical properties (e.g., tensile strength) is directly related to molecular weight reduction.

The time-dependent behaviour of biodegradable PLA-PCL fibres during their hydrolytic degradation in a phosphate buffer medium has been studied by Viera et al. [152,153] (37 °C, 16 weeks). It was found that the decrease of tensile strength σ of the fibres follows the same trend as the decrease of molecular weight *M_n_*. Modelling hydrolysis as a first-order kinetic mechanism, the hydrolytic damage $d_{h}$ is defined as follows [152]:(21)$$d_{h}=1-\frac{\sigma_{t}}{\sigma_{0}}=1-\frac{M_{nt}}{M_{n0}}=1-e^{-ut}$$
where *u* is is the degradation rate of the material. Subscripts *t* and 0 are related to the corresponding parameters at current time t and initial (non-degraded) values. By incorporating Equation (21) into the constitutive models such as Neo-Hookean and Mooney–Rivlin hyperelastic models [152] and Bergström–Boyce viscoelastic model for polymer undergoing large deformations [153], stress–strain behaviour of PLA-PCL fibres for different degradation times was effectively predicted. The approach could be extended to other biodegradable polymers.

Singh and coauthors have studied the effect of hydrolytic degradation and strain rate on the tensile properties of PLA fibres [156]. The authors applied the modified three-element standard solid model (according to Khan [157]) introducing degradation-dependent stiffness parameters, while viscous (strain-dependent) parameters are assumed to be unaffected by degradation. Stiffness (Young modulus *E*) degradation was presented in a similar to Equation (21) way by using an exponential law:(22)$$\frac{E_{t}}{E_{0}}=e^{-ut}$$

The model Equation (22) parameters are determined experimentally by assessing *E* changes upon different degradation times.

Breche and coauthors have studied the evolution of the physical and mechanical properties of PLA-b-PEG-b-PLA biodegradable triblock copolymers caused by hydrolytic degradation [154]. Samples were immersed in phosphate buffer solution at 37 °C for up to 12 weeks. The stress relaxation was modelled by a linear viscoelastic model introducing a stiffness-related degradation variable. Similarly to [156], the viscous component was considered independent of degradation time, at least at the early stages. Then, following Viera et al.’s definition [152] and Equation (21), the degradation variable is defined as
(23)$$d_{h}=1-\frac{\sigma_{max}\left(d_{h}\right)}{\sigma_{max}\left(d_{h}=0\right)}=1-\frac{\sigma_{relax}\left(d_{h},0\right)}{\sigma_{relax}\left(d_{h}=0,0\right)}$$
where $\sigma_{max}$ is the maximum stress reached at the end of load for a given degradation state $d_{h}$ and corresponds to the time zero of relaxation, while $\sigma_{relax}$ is the initial value of the relaxation stress for degraded and undegraded material. The degradation variable Equation (23) was linked to the molecular weight changes of the material according to the following dependencies [153]:(24)$$\begin{array}{ccc} d_{h}=c\cdot \left(1-\frac{M_{nt}}{M_{n0}}\right), & & t<t_{c}\\ d_{h}=a\cdot exp\left(-b\frac{M_{nt}}{M_{n0}}\right), & & t\geq t_{c} \end{array}$$
where *a*, *b* and *c* are material parameters; $t_{c}$ is a critical time, when the evolution of properties degradation is changed considerably (3 weeks for the PLA-b-PEG-b-PLA copolymers understudy).

Zhang et al. [158] proposed a semiempirical approach for predicting the strength of biodegradable medical polyesters, namely, PLA and polyglycolide (PGA) and their copolymers, during hydrolytic degradation. Three different phases in the mesoscopic-scale (amorphous, crystalline and vacancy phases) were defined and further integrated into the multiscale heterogeneous strength model. The strength of the amorphous and crystalline phases was related to molecular weight through power-law dependencies, while the cavity-related zones were considered as the zero-strength phase. The total strength is given by the equation:(25)$$\sigma \left(t\right)=\sigma_{A0}\cdot \alpha_{A}{\left[\frac{M_{nA}\left(t\right)}{M_{n0}}\right]}^{\beta_{A}}+\sigma_{C0}\cdot \alpha_{C}{\left[\frac{M_{nC}\left(t\right)}{M_{n0}}\right]}^{\beta_{C}}$$
where $\sigma_{0}$ and $M_{n0}$ are the initial strength and molecular weight, respectively. $M_{n}\left(t\right)$ is the molecular weight at time *t*, α and β are material parameters. Subscripts *A* and *C* in Equation (25) are related to the amorphous and crystalline phases, respectively.

The considered models and methods may be adapted and used for other biodegradable polymers and nanocomposites. The incorporation of nanoparticles into BP contributed to the improved barrier properties and decreased degradation of polymers [13,14,111,141] that will appear in the decreased damage parameter *d*_h_ in Equations (21)–(25).

The models could be considered effective tools in designing biopolymer composites with tailored degradation and durability.

## 6. Conclusions

Environmental degradation mainly promotes a significant decrease in mechanical properties, particularly when the molecular weight of BP is low. Thus, incorporating nanofillers into BP could attenuate the loss of mechanical properties and improve durability. At the same time, in the case of poor dispersion, the nanofillers can lead to faster degradation, depending on the structural integrity and the state of interfacial adhesion. To fully understand and interrelate numerous factors (e.g., moisture, temperature, etc.) that can affect the degradation process of BPN, combined and comprehensive scientific investigations are required.

A correlation between outdoor and accelerated weathering should be established experimentally and particular analytical models should be developed. The degradation of mechanical properties could be modelled as a function of the duration of environmental ageing. The incorporation of nanoparticles into BP contributed to the improved barrier properties and decreased degradation of BP. The models considered in the review could be effective tools in designing biopolymer composites with tailored degradation and durability. Moreover, models could be developed to combine the effects of temperature and humidity to predict the durability of BP and BPN.

## Figures and Tables

**Figure 1 polymers-13-03375-f001:**
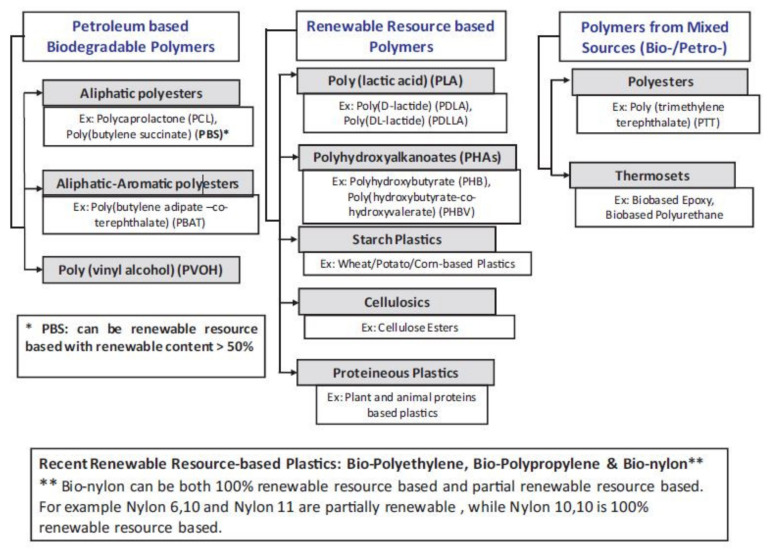
Classification of biopolymers. Reproduced with permission from [12]. Copyright © (2013). Elsevier Ltd. (licence No. 5142931044479).

**Figure 2 polymers-13-03375-f002:**
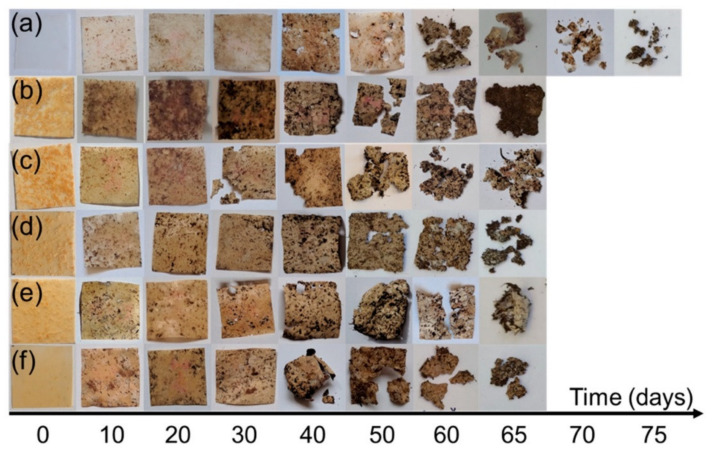
Photos of (**a**) PBS; (**b**) 40% NFC; (**c**) 7/3; (**d**) 5/5; (**e**) 3/7 and (**f**) 40% MCC films during biodegradation studies in soil burial test conducted in composting conditions [55].

**Figure 3 polymers-13-03375-f003:**
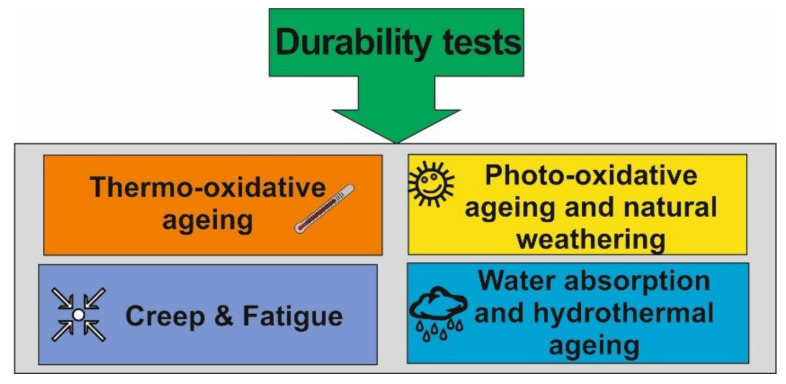
Basic types of durability tests for polymer composite materials.

**Figure 4 polymers-13-03375-f004:**
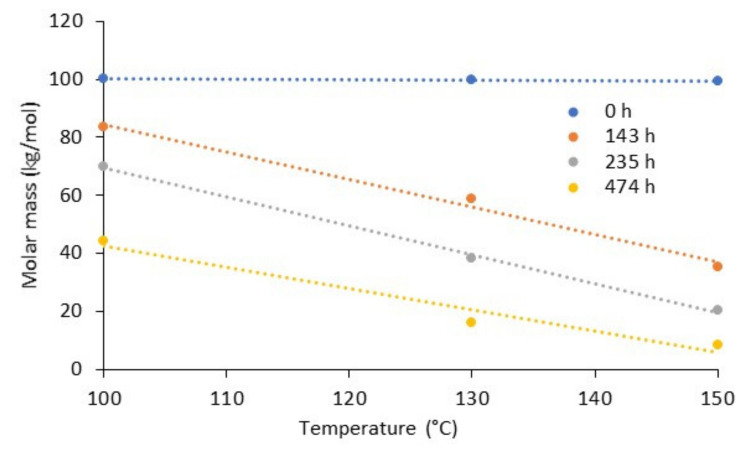
Molar weight of PLA during thermo-oxidative ageing as a function of exposure temperature. Dots: experimental data used from [116]; lines: linear approximations.

**Figure 5 polymers-13-03375-f005:**
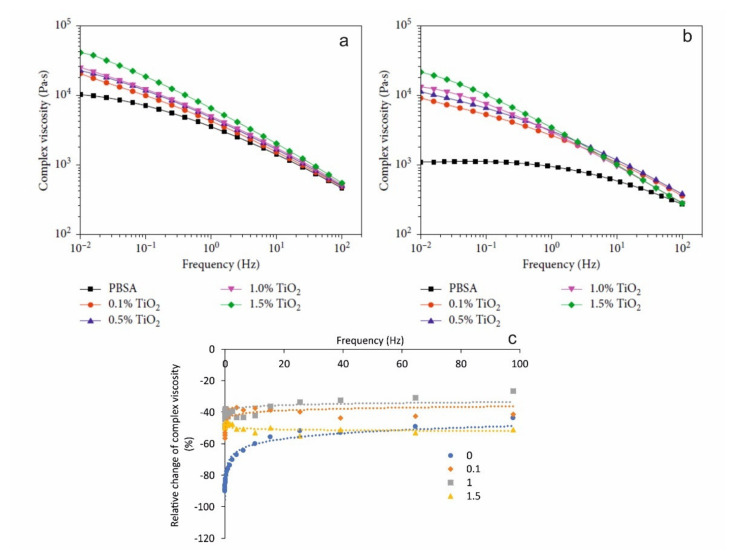
Complex viscosity of pure PBSA and PBSA/TiO_2_ nanocomposites (**a**) before and (**b**) after 360 h of UV irradiation as a function of frequency at 140 °C (reproduced from [100], copyright © (2019). Hindawi, (**c**) relative change of complex viscosity of PBSA filled with TiO_2_ (relative weight content is indicated on the graph) vs. frequency. Dots: experimental data used from [100]; lines: approximations by logarithmic functions.

**Figure 6 polymers-13-03375-f006:**
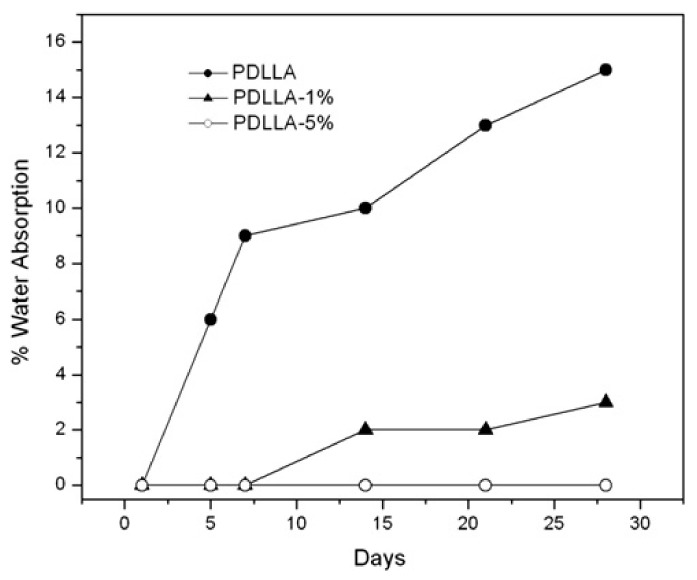
Kinetics of water absorption of poly(D,L-lactide) filled with cellulose nanowhiskers at different filler contents indicated on the graph. Reproduced with permission from [13]. Copyright © (2011). Elsevier Ltd. (licence No. 5135221438116).

**Figure 7 polymers-13-03375-f007:**
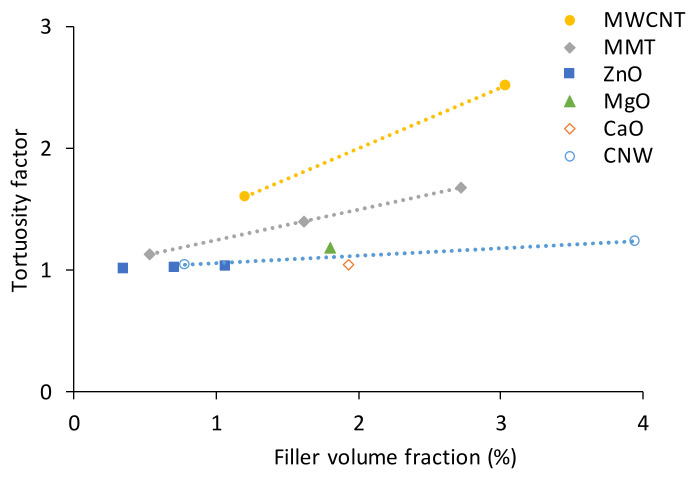
Tortuosity factor as a function of filler volume fraction for PLA filled with different nanofillers (indicated in the legend).

**Figure 8 polymers-13-03375-f008:**
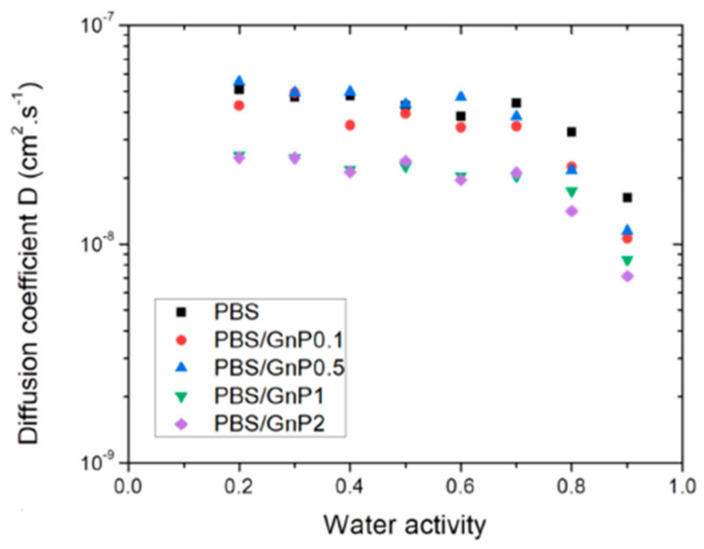
Diffusion coefficient as a function of water activity for neat PBS and PBS/GnP nanocomposites [99].

**Figure 9 polymers-13-03375-f009:**
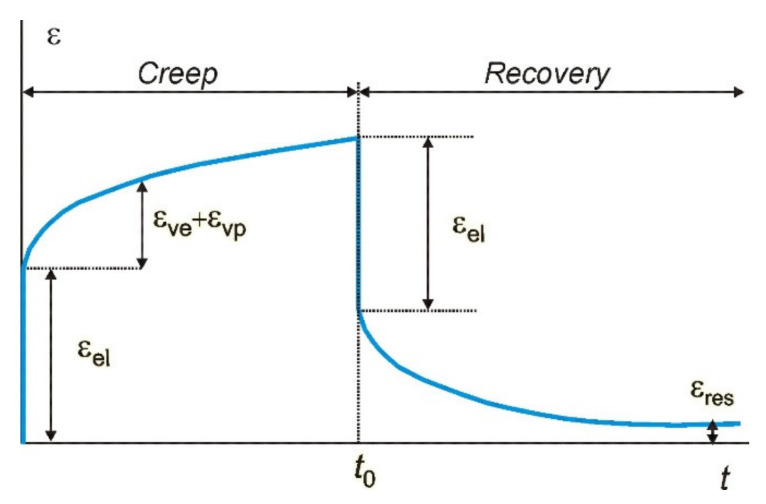
A schematic strain vs. time curve in a creep-recovery test.

**Figure 10 polymers-13-03375-f010:**
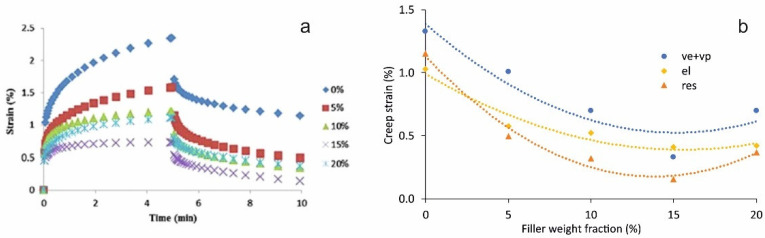
The (**a**) creep-recovery vs. time curves of starch only and starch–CNF composite films (reproduced with permission from [10], copyright © (2015), Elsevier Ltd., licence No. 5143001008321). (**b**) Viscoelastic and viscoplastic (ve+vp), elastic (el) and residual (res) strains of starch modified with cellulose nanofibrils vs. filler weight fraction. Dots: experimental data from [10]; lines: approximations by polynomial functions.

**Table 1 polymers-13-03375-t001:** BP and their characteristic physical and mechanical properties [35,36,37,38,39,40].

	PLA	PCL	PBS	PBAT	PHA	TPS
Density, g/cm^3^	1.21–1.30	1.11–1.15	1.22–1.26	1.26	1.18–1.26	0.85–1.00
Melting point, °C	165–170	58–65	110–115	89	160–190	100–160 *
Glass transition, °C	55–65	−65–60	−35–20	−30–20	10–40	−60–10
Tensile strength, MPa	30–60	20–45	20–35	15–25	30–50	0.5–50
Young’s modulus, GPa	2–4	0.2–0.4	0.2–0.4	0.05–0.10	3–4	0.05–0.50
Elongation at break, %	2–10	300–1000	30–500	500–1100	4–12	10–300

* TPS does not melt but is processed at these temperatures.

**Table 2 polymers-13-03375-t002:** Scope of applications for BPN.

Type of Application	Biopolymer	Nanofiller	References
Packaging	PLA	ZnO	[11,48,49,50,51]
	PLA	MMT	[17,49]
	PLA	Nanocellulose	[48,51,52,53]
	PBS	ZnO	[54]
	PBS	Nanocellulose	[48,55]
	Starch	Ag, ZnO, CuO	[56]
	Starch	Nanocellulose	[48]
	PCL	ZnO/nanocellulose	[57]
Biomedical applications	PLA	ZnO	[26]
	PLA	TiO_2_	[58]
	PLA	Fe_3_O_4_	[47]
Antimicrobial applications	PLA	Ag	[49]
	PLA	MMT	[1,17]
	PBS	ZnO	[16,17]
	Cellulose acetate	Cu	[59]
Smart applications	PLA	MWCNT	[18]
	Poly(d,l-lactide)	Fe_3_O_4_	[19]
	PLA/PHBV	MWCNT	[20]

**Table 3 polymers-13-03375-t003:** Recent studies on durability of various BPN *.

BP Matrix	Filler (Content)	Type of Durability Testing	Indicator	Reference
PLA	ZnO (0.1, 1 wt.%)	Thermal	Glass trans. temperature	[26]
	ZnO (1, 2, 3 wt%)	Water absorption	Diffusivity	[11]
	CaO, MgO (5 wt%)	Thermal	Pyrolysis	[47]
	MMT (5 wt.%)	Microbial	Molecular weight	[17]
	CNT (2, 5 wt%)	Thermal	Crystal. temperature	[15]
	CNF (1, 5 wt.%)	Hydrothermal	Glass trans. temperature	[13]
	CNC (1, 5 wt.%)	Water absorption	Hydrolytic degradation rate	[96]
	ZnO: Cu/Ag (0.5–1.5 wt%)	Microbial	SEM images	[50]
	Nanoclays (OMMT, HNT, Laponite^®^, 1, 5 wt.%)	Microbial	CO_2_ evolution	[97]
	SiO_2_	Creep tests	Creep resistance	[98]
PBS	ZnO (0.5, 1, 3 wt.%)	Photo-oxidative	Crystal. temperature	[54]
	ZnO (2–10 wt.%)	Microbial	Inhibition zone diameter	[16]
	MMT (0–10 wt.%)	Hydrothermal	Tensile strength and modulus	[95]
	GnP	Water absorption	Permeability	[99]
	CNF (12–40 wt.%)	Thermal	Crystal. temperature	[55]
PBSA	TiO_2_ (0.5–1.5 wt.%)	Photo-oxidative	Crystal. temperature	[100]
Starch	CNF (5–20 wt.%)	Thermal	Creep resistance	[10]
	Ag, ZnO, CuO (0.66–3 wt%)	Microbial	SEM images	[56]
	Cellulose nanofibres (10 wt.%)	Thermal	Activation energy	[101]
	MWCNT (0.005–0.055 wt%))	Thermal	Glass trans. temperature	[102]
PHB	Bentonite (2–6 wt.%)	Thermal	Crystal. temperature	[103]
	nAg (0.25–1.25 mM)	Microbial, hydrolytic	SEM, glass trans. temperature	[104]
	MMT (1–10 wt.%)	Thermal	Glass trans. temperature	[87]
PCL	Nanoclay (6–26 wt.%)	Thermal	Glass trans. temperature	[105]
	Nanocellulose/ZnO (2–8 wt.%)	Thermal	Phase trans. temperature	[57]
	Bentonite (1.5, 3 wt.%)	Creep	Creep resistance	[22]
	MMT, MWCNT, SiO_2_ (0.5–2.5 wt.%)	Thermal	Activation energy	[106]
	GO (0.1 wt%)	Creep	Creep resistance	[107]
Cellulose acetate	Cu (2, 6 mol.%)	Microbial	SEM images	[59]
	Ag/MMT (3, 5 wt.%)	Microbial, thermal	Inhibition reduction rate, glass trans. temperature	[86]
PVA	CNC/GO/Ag (0.5 wt.%)	Bacterial	Antibacterial efficiency	[108]
PLA/PHBV	TiO_2_	Thermal	Activation energy	[109]
PLA/PBS	CNC (1–3wt.%)	Barrier	Permeability, oxygen transmission rate	[96]
PBAT	CNT (1–5wt.%)	Creep and stress relaxation	Creep resistance	[110]
PVA/ST/GL	HN (0.25–5 wt.%)	Water absorption	Water solubility, water contact angle	[111]

* designations according to the list of abbreviations.

## Data Availability

Not applicable.

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
