# Peer review of "Durability of Biodegradable Polymer Nanocomposites"

_polymers, 2021, doi:10.3390/polym13193375_

Round 1
Reviewer 1 Report
Biodegradable polymers are the future of the packaging industry. All modifications influencing the improvement of mechanical, aesthetic and functional properties are very desirable. When using all kinds of additives, you should remember about the requirements that are placed on food and cosmetics packaging. There is no such information in the presented article.
- The introduction to work is interesting.
- The measurement methodology and the presentation of the results are correct.
- Figures are legible and well described.
- The conclusions are correctly formulated.
- The amount of literature cited is very large. It would be worth limiting it to the most important items.
Author Response
Biodegradable polymers are the future of the packaging industry. All modifications influencing the improvement of mechanical, aesthetic and functional properties are very desirable. When using all kinds of additives, you should remember about the requirements that are placed on food and cosmetics packaging. There is no such information in the presented article.
Dear Reviewer,
Thank you for your valuable comments. We agree that special requirements are placed on food and cosmetics packaging. For sure, this information was provided in the references regarding packaging applications. The purpose of our paper was to review and discuss current literature results received on the durability of biodegradable polymer nanocomposites considering the long-term aspects of their applications in the future. It should be mentioned that the durability of biodegradable polymers and composites has become an important aspect, especially under environmental aging which restricts this new class of sustainable materials for advanced applications. Effects induced by the addition of nanofillers are not strictly linked with their chemical composition and often, there are more expensive alternatives suitable for food and cosmetic packaging.
The introduction to work is interesting.
The measurement methodology and the presentation of the results are correct.
Figures are legible and well described.
The conclusions are correctly formulated.
The amount of literature cited is very large. It would be worth limiting it to the most important items.
Since it is a review paper the effors were being made to review the most important items available in the literature on the recent results of the durability of biodegradable polymers and polymer nanocomposites, their basic engineering properties, potential nanofillers to improve the performance of these materials, biodegradation aspects, and modelling of mechanical properties accompanied by biodegradation.
We agree that the amount of literature is large but the topic we discussed is relatively wide and includes many important aspects. Since we tried to include the recent experimental results and the modeling of mechanical properties, the literature analysis should provide a comprehensive overview of the durability of biodegradable polymers and polymer nanocomposites.
Reviewer 2 Report
Dear Author
Plagiarism was 5%
the article entitled "Durability of biodegradable polymer nanocomposites" which was submitted to polymers for publication is written well and explained all the data with excellent behaviour therefore it is suitable to be published in polymers
why author didn't discuss the important biodegradable polysaccharide polymers such as cyclodextrin, chitin, pectin and so on in this review article

Author Response
Plagiarism was 5%
the article entitled "Durability of biodegradable polymer nanocomposites" which was submitted to polymers for publication is written well and explained all the data with excellent behaviour therefore it is suitable to be published in polymers
why author didn't discuss the important biodegradable polysaccharide polymers such as cyclodextrin, chitin, pectin and so on in this review article
Dear Reviewer,
We highly appreciate your comments. We have modified the abstract to reflect the selection of biodegradable polymers better. We chose these polymer materials considering their ability to replace commodity plastics. This includes prices, resistance to water, high mechanical properties, good processability, etc. The materials that meet these highly competitive conditions are limited, thus the authors chose to focus on biodegradable polyesters (like PCL, PLA, PBS, PBAT) and cheap, abundant biopolymers like cellulose and starch that can be easily modified. We considered the market report when selecting topical biodegradable polymers for review https://www.ifbb-hannover.de/en/facts-and-statistics.html .